# Revisiting Sparse Retrieval for Few-shot Entity Linking

**Yulin Chen**[*], **Zhenran Xu**[*], **Baotian Hu**[†] and **Min Zhang**
Harbin Institute of Technology (Shenzhen), Shenzhen, China
{200110528, xuzhenran}@stu.hit.edu.cn
{hubaotian, zhangmin2021}@hit.edu.cn

## Abstract

Entity linking aims to link ambiguous mentions to their corresponding entities in a knowledge base. One of the key challenges comes from insufficient labeled data for specific domains. Although dense retrievers have achieved excellent performance on several benchmarks, their performance decreases significantly when only a limited amount of in-domain labeled data is available. In such few-shot setting, we revisit the sparse retrieval method, and propose an ELECTRA-based keyword extractor to denoise the mention context and construct a better query expression. For training the extractor, we propose a distant supervision method to automatically generate training data based on overlapping tokens between mention contexts and entity descriptions. Experimental results on the ZESHEL dataset demonstrate that the proposed method outperforms state-of-the-art models by a significant margin across all test domains, showing the effectiveness of keyword-enhanced sparse retrieval. Code is available at https://github.com/HITsz-TMG/Sparse-Retrieval-Fewshot-EL.

## 1 Introduction

Entity linking (EL) aligns entity mentions in documents with the corresponding entities in a knowledge base (KB), which is a crucial component of information extraction (Ji and Nothman, 2016). Typically, EL systems follow a "retrieve and rerank" pipeline (Logeswaran et al., 2019): Candidate Retrieval, where a small set of candidates are efficiently retrieved from a large number of entities, and Candidate Ranking, where candidates are ranked to find the most probable one. With the rapid development of pre-trained language models (PLM), EL has witnessed a radical paradigm shift towards dense retrieval (Ma et al., 2021; Sun et al.,

2022). Using bi-encoders and Approximate Nearest Neighbors (ANN) search has become the standard approach for initial entity retrieval, overcoming the long-standing vocabulary mismatch problem and showing promising performance gains.

Although the bi-encoder has shown strong in-domain performance, they typically require training on sufficient labeled data to perform well in a specific domain (Ren et al., 2022). However, such labeled data may be limited or expensive in new and specialized domains. As reported by Li et al. (2022b), in the absence of sufficient training data, the recall of bi-encoder significantly decreases, performing even worse than unsupervised sparse retrievers (e.g., BM25 (Robertson and Zaragoza, 2009)). Motivated by the above finding, in this work, instead of resorting to dense representations in the "semantic space", we go back to the "lexical space" and explore PLM-augmented sparse representations for few-shot EL.

When applying BM25 to the first-stage retrieval, previous work only employs the mention string as query (Wu et al., 2020). However, this can be insufficient due to under-specified mentions (e.g., "his embodiments" in Figure 1), requiring additional context to formulate a more comprehensive query. While an intuitive solution could be incorporating all context into the query, this approach is suboptimal for two reasons: (i) it does not create mention-specific queries, i.e., for all mentions in a document, their queries and retrieved entities are the same; (ii) it may introduce noise into queries (Mao et al., 2022), since not all words in the context are necessary for understanding the mention.

This paper introduces a keyword extractor to denoise the context for BM25 retrieval. The extraction is modeled as binary token classification, identifying which tokens in context should be added to the mention string. We employ a discriminative PLM, specifically ELECTRA (Clark et al., 2020), as the basis of our keyword extractor, and leverage

---

[*]Both authors contributed equally to this work.
[†]Corresponding author.

its discriminator head to predict a score for each token. The top-$k$ tokens with the highest scores are then added into the final query.

Training the extractor requires binary labels to distinguish whether a token is a keyword. The *keywords* are expected to be related to the mention and necessary for understanding it. We propose a distant supervision method to automatically generate training data, based on overlapping words between the mention context and the description of the corresponding entity. These overlapping words are then ranked based on their BM25 score, which measures the relevance of the word to the entity description. We select the top-$k$ words with the highest scores as *keywords*.

Following the few-shot experimental setting in Li et al. (2022b), We evaluate our approach on the ZESHEL dataset (Logeswaran et al., 2019). The results highlight our approach's superior performance across all test domains, with an average of 15.07% recall@64 improvement over the best dense retriever result. In addition to its superior performance, our method inherits the desirable properties of sparse representations such as efficiency of inverted indexes and interpretability.

The contributions of this work are threefold:

- We are the first to explore PLM-augmented sparse retrieval in entity linking, especially in scenarios of insufficient data.

- We propose a keyword extractor to denoise the mention context, and use the keywords to formulate the query for sparse retrieval, accompanied by a distant supervision method to generate training data.

- We achieve the state-of-the-art performance across all test domains of ZESHEL, outperforming previous dense retrieval methods by a large margin.

## 2  Related work

Entity linking (EL) bridges the gap between knowledge and downstream tasks (Wang et al., 2023; Dong et al., 2022; Li et al., 2022a). Recent entity linking (EL) methods follow a two-stage "retrieve and re-rank" approach (Wu et al., 2020), and our work focuses on the first-stage retrieval. Traditional retrieval systems based on lexical matching struggle with vocabulary mismatch issues (Formal et al., 2021). With the advent of pre-trained language models, the bi-encoder has largely mitigated this problem, but its effectiveness depends on extensive annotation data (Ren et al., 2022). Without sufficient labeled data in specific domains, the recall of the bi-encoder drastically decreases (Li et al., 2022b), hence the need for further research about few-shot EL.

*Few-shot Entity Linking* is recently proposed by Li et al. (2022b). They split the data in a specific domain of ZESHEL (Logeswaran et al., 2019) and provide few examples to train. They address the data scarcity problem with synthetic mention-entity pairs, and design a meta-learning mechanism to assign different weights to synthetic data. With this training strategy, the performance of bi-encoder is better than simply training with insufficient data. However, it still cannot surpass the performance of the unsupervised sparse retrieval method (i.e., BM25 in Table 2). As dense retrievers (e.g., the bi-encoder) and generative retrievers (e.g., GENRE (Cao et al., 2021)) are data-hungry, we go back to the "lexical space" and resort to sparse retrievers for few-shot EL solutions.

## 3  Methodology

When applying BM25 to the first-stage retrieval of EL, previous work directly uses mention string as query. We propose a keyword extractor to denoise the context, and add the extracted keywords into the query. Then the expanded query is used for BM25 retrieval. In Section 3.1, we introduce our distant supervision method to generate keyword labels. In Section 3.2, we present the architecture, training and inference of our keyword extractor.

**Preprocessing.** BM25 implementation generally comes with stopword removal (Yang et al., 2017). In this work, we remove the words which occur in more than 20% of entities' description documents, resulting in removal of uninformative words (such as "we", "the", etc.).

### 3.1  Distant Supervision

Given a mention $m$, its context $M$ and the description document $E$ of its corresponding entity, the distant supervision method aims to gather keywords in the context $M$ related to the description $E$. As shown in Figure 1(a), the overlapping words form a preliminary keyword set (denoted as $\mathcal{K}^* = \{w_1, w_2, \ldots, w_{|\mathcal{K}^*|}\}$), i.e.,

$$\mathcal{K}^* = words(M) \cap words(E) \qquad (1)$$

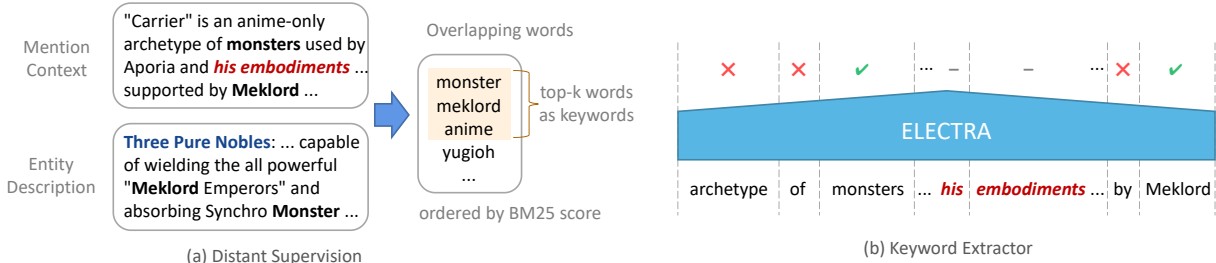

Figure 1: Overview of our method. The distant supervision method (left) generates weak keyword labels based on overlapping words. Then the keyword extractor (right) is trained on these generated keyword labels.

where $words(x)$ is the set of words in sequence $x$. The words in $\mathcal{K}^*$ are further ranked according to the BM25 score between each word $w_i$ and the document $E$, calculated as follows:

$$s(w_i, E) = \frac{\text{IDF}(w_i) \cdot f(w_i, E) \cdot (k_1 + 1)}{f(w_i, E) + k_1 \cdot (1 - b + b \cdot \frac{|E|}{\text{avgdl}})} \quad (2)$$

where $\text{IDF}(w_i)$ represents the inverse document frequency of word $w_i$, $f(w_i, E)$ denotes the frequency of word $w_i$ appearing in description $E$, $|E|$ is the length of the description $D$, avgdl is the average document length in entity descriptions. The hyperparameters $k_1$ and $b$ are adjustable values, where we set $k_1 = 1.5$ and $b = 0.75$. We select the top-$k$ words with the highest scores as the final keyword set $\mathcal{K}$ for the mention $m$.

### 3.2 Keyword Extractor

Our keyword extractor aims to denoise the context through sequence labeling. Since discriminative pre-trained models can be easily adapted to tasks involving multi-token classification (Xia et al., 2022), we use ELECTRA as the backbone of our keyword extractor, as shown in Figure 1(b). For the mention $m$ in its context $M$, the input $\tau_m$ is the word-pieces of the highlighted mention and its context:

$$\texttt{[CLS]}\ M_l\ \texttt{[START]}\ m\ \texttt{[END]}\ M_r\ \texttt{[SEP]}$$

where $M_l$ and $M_r$ are context before and after the mention $m$ respectively. $\texttt{[START]}$ and $\texttt{[END]}$ are special tokens to tag the mention. The token embeddings from the last layer of the extractor $T$ can be denoted as follows:

$$\boldsymbol{H} = T(\tau_m) \in \mathbb{R}^{L \times d} \quad (3)$$

where $L$ is the number of word-pieces in $\tau_m$ and $d$ is the hidden dimension. Finally, the predicted score of the $i$-th token, denoted as $\hat{s}_i$, is calculated

| Domains | #Train | #Dev | #Test |
|---|---|---|---|
| Forgotten Realms | 50 | 50 | 1100 |
| Lego | 50 | 50 | 1099 |
| Star Trek | 50 | 50 | 4127 |
| YuGiOh | 50 | 50 | 3274 |

Table 1: Statistics of the few-shot entity linking dataset ZESHEL.

with a discriminator head (i.e., a linear layer $\boldsymbol{W}$) and the sigmoid activation function:

$$\hat{s}_i = \text{Sigmoid}(\boldsymbol{h}_i \boldsymbol{W}) \quad (4)$$

where $\boldsymbol{h}_i \in \mathbb{R}^{1 \times d}$ is $i$-th row of the matrix $\boldsymbol{H}$. $\hat{s}_i \in [0.0, 1.0]$, and a higher score means the token is more likely to be a keyword.

**Optimization.** In Section 3.1, for each mention $m$ and its context $M$, we have obtained the keyword set $\mathcal{K}$. The tokens of every word in $\mathcal{K}$ should be labeled as keywords. We optimize the extractor with a binary cross entropy loss, as below:

$$\mathcal{L} = -\frac{1}{L} \sum_{i=1}^{L} s_i \log(\hat{s}_i) + (1 - s_i) \log(1 - \hat{s}_i) \quad (5)$$

where $s_i$ takes the value 1 if the $i$-th token should be labeled as a keyword, otherwise 0.

**Inference.** The keyword extractor predicts a score $\hat{s}_i$ for every token. The top-$k$ distinct tokens with the highest scores are selected for the keyword set $\hat{\mathcal{K}}$. The final query for BM25 combines the extracted keywords with the mention $m$:

$$\mathcal{Q} = words(m) \cup \hat{\mathcal{K}} \quad (6)$$

The top-$n$ retrieved entities by BM25 are the result of our method.

## 4 Experiment

### 4.1 Dataset and Evaluation Metric

For fair comparison with Li et al. (2022b), we follow their few-shot experimental settings, i.e.,

| Method | Forgotten Realms | Lego | Star Trek | YuGiOh | Macro Average | Micro Average |
|---|---|---|---|---|---|---|
| BLINK (Wu et al., 2020) | 35.27 | 52.68 | 21.57 | 35.00 | 36.13 | 31.28 |
| MUVER (Ma et al., 2021) | 37.27 | 66.15 | 23.89 | 56.38 | 45.92 | 41.34 |
| GENRE (Cao et al., 2021) | 67.27 | 50.32 | 55.75 | 29.14 | 50.62 | 47.37 |
| DL4EL (Le and Titov, 2019) | 66.71 | 75.12 | 59.91 | 59.20 | 65.24 | 62.19 |
| MetaBINK (Li et al., 2022b) | 69.56 | 78.17 | 61.41 | 61.28 | 67.61 | 64.22 |
| BM25 *(mention words)* | 83.87 | 81.62 | 65.99 | 61.03 | 73.13 | 68.14 |
| BM25 *(mention context)* | 75.22 | 74.19 | 59.20 | 71.45 | 70.02 | 66.93 |
| Ours | **88.53** | **85.48** | **74.81** | **79.75** | **82.14** | **79.29** |
| Ours (w/o mention words) | 80.36 | 77.53 | 62.03 | 74.40 | 73.58 | 70.12 |

Table 2: Recall@64 of our method compared with dense and generative retrievers (top) and sparse retrievers (bottom) on all test domains of ZESHEL. **Bold** denotes the best results.

| Mention Context | Corresponding Entity Description |
|---|---|
| Birth of the Ultimate Tag is the forty seventh chapter of the Yu Gi Oh GX manga. It was first printed in Japanese in the V Jump magazine and in English in the Shonen Jump magazine. Both of which were printed in ***volume 7*** of the Yu Gi Oh GX graphic novels afterwards … | Yu-Gi-Oh! GX-Volume 7 King Atticus True Power!!, known as The King ' s True Power!! in the Japanese version, is the seventh volume of the Yu -Gi-Oh! GX manga. |
| Anzu bought magazines with tourist information. The two met at Domino Station at 10:00 where Anzu was nervous at first. They first went to ***a coffee shop*** for drinks. … | Domino Coffee is a restaurant in Domino City in the anime and manga. …Gardner and Yami Yugi went here among other places on their date. … |

Table 3: Examples of the extracted keywords and their overlap with the corresponding entity descriptions. ***Italic*** denotes mentions. red in contexts shows the extracted keywords. Red in entity descriptions represents words overlapping with the keyword set.

choose ZESHEL (Logeswaran et al., 2019) as the dataset, split the data as shown in Table 1 and use top-64 recall (i.e., Recall@64) as the evaluation metric.

### 4.2 Baselines

To evaluate our method, we compare it with the state-of-the-art sparse retriever (i.e., BM25 (Robertson and Zaragoza, 2009)), dense retrievers (i.e., BLINK (Wu et al., 2020), MUVER (Ma et al., 2021), MetaBLINK (Li et al., 2022b), DL4EL (Le and Titov, 2019)), and the generative retriever GENRE (Cao et al., 2021). More details of the baselines can be found in Appendix A. Implementation details are in Appendix B.

### 4.3 Results

**Main results.** Table 2 shows that our model outperforms all dense and generative retrieval methods across all domains, and achieves a 15.07% micro-averaged Recall@64 improvement over the best reported result (i.e., MetaBLINK).

**Comparison among sparse retrievers** in Table 2 highlights the importance of mention-related context. BM25 *(mention words)* is apparently not the optimal solution since it ignores all context; BM25 *(mention context)* leverages all context and

performs even worse, indicating the importance of context denoising. The YuGiOh domain has the lowest recall of BM25 *(mention words)*, showing a large vocabulary mismatch between mention string and entity documents. However, with mention-related context, our method addresses this issue effectively, resulting in an impressive 18.72% performance improvement in the YuGiOh domain.

**Ablation study** in Table 2, i.e. results of BM25 *(mention words)* and Ours (w/o mention words), demonstrates that the mention string and the keywords are both important to the final query. Deleting either of them in BM25's query will result in a significant performance drop. To further explore the contribution of keywords, we present the **case study** in Table 3. From these cases, we can find that the extracted keywords are semantically related to the mention, excluding the noise in context.

## 5 Conclusion

In this work, we focus on the first-stage retrieval in few-shot entity linking (EL). Given the significant performance drop in dense retrievers for few-shot scenarios, we revisit the sparse retrieval method and enhance its performance with a keyword extractor. We also propose a distant supervision ap-

proach for automated training data generation. Our extensive experiments demonstrate our method's superior performance, with case studies showing denoised context and mention-related keywords. Future work may focus on more PLM-augmented sparse retrieval solutions for EL, expanding the query to words outside the context.

## Limitations

Despite surpassing the previous state-of-the-art performance by a significant margin, our method exhibits certain limitations, primarily related to the constraint on the keywords. The formulated query effectively denoises the context, but is still limited to the scope of mention context. The vocabulary mismatch between the mention context and the entity description cannot be alleviated in our method. Expanding the query to words outside the context could be an interesting research direction.

## Acknowledgments

We thank Zifei Shan for discussions and the valuable feedback. This work is jointly supported by grants: Natural Science Foundation of China (No. 62006061, 82171475), Strategic Emerging Industry Development Special Funds of Shenzhen (No.JCYJ20200109113403826).

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

## A  Baseline details

To evaluate the performance of our method, we compare with the following state-of-the-art retrievers that represent a diverse array of approaches.

- BM25 (Robertson and Zaragoza, 2009), a traditional and strong sparse retrieval method. We consider two variants: *BM25 (mention words)*, which simply uses the mention string as query, and *BM25 (mention context)*, which uses all context as query.

- BLINK (Wu et al., 2020) adopts the bi-encoder architecture for dense retrieval.

- MUVER (Ma et al., 2021) extends the bi-encoder to multi-view entity representations and approximates the optimal view for mentions via a heuristic searching method.

- GENRE (Cao et al., 2021) is an autoregressive retrieval method base on generating the title of candidates with constrained beam search.

- MetaBLINK (Li et al., 2022b) proposes a weak supervision method to generate mention-entity pairs. These pairs are then used for training BLINK with a meta-learning mechanism.

- DL4EL (Le and Titov, 2019) is a denoising method based on KL divergence to force the model to select high-quantity data. It can be combined with MetaBLINK's generated data to train BLINK.

## B  Implementation Details

Our model is implemented with PyTorch 1.10.0. The number of the model parameters is roughly 109M. All experiments are carried out on a single NVIDIA A100 GPU. We use Adam optimizer (Kingma and Ba, 2015) with weight decay set to 0.01, learning rate set to 2e-5. The batch size is 8, and the number of extracted keywords (denoted as $k$) is 32, For the convenience of comparison, the number of final retrieved entities (denoted as $n$) is 64. The mention context length is set to 128, with each left and right context having a length of 64. For each domain, we train for a total of 10 epochs, and choose the best checkpoint based on the performance of development set. The whole training process takes about 1.5 minutes for each domain.

In Table 2, the results of MUVER and GENRE are our reproduction based on their official Github repositories. These reproduction results, together with results of sparse retrievers, are from 5 runs of our implementation with different random seeds.