# OpenReview forum: "Revisiting Sparse Retrieval for Few-shot Entity Linking"
_EMNLP/2023/Conference — EMNLP 2023 Main_

### Official Review · Reviewer_paUf · 2023-08-02

**Soundness:** 4

**Excitement:**

4: Strong: This paper deepens the understanding of some phenomenon or lowers the barriers to an existing research direction.

**Paper Topic And Main Contributions:**

This work addresses the retrieval of candidates problem in few-shot entity linking task. It offers a PLM based keyword extractor to construct a refined context and uses BM25 retrieval model to obtain the candidate entities. Experimental results validate that the proposed model can achieve state-of-the-art performance.

**Reasons To Accept:**

S1. The paper is well written and clearly organized.

S2. The motivation is clear and the claim is well supported.


**Reasons To Reject:**

W1. Some concepts need to be explained to ease the reading. For instance, it is unclear what is the bi-encoder and why it depends on the training data. Besides, it is unclear what ELECTRA is and how it works.

W2. Is the proposed model also effective on normal EL datasets?


**Reproducibility:**

4: Could mostly reproduce the results, but there may be some variation because of sample variance or minor variations in their interpretation of the protocol or method.

**Reviewer Confidence:**

3: Pretty sure, but there's a chance I missed something. Although I have a good feel for this area in general, I did not carefully check the paper's details, e.g., the math, experimental design, or novelty.

---

> ### Author Rebuttal · Authors · 2023-08-29
>
> Thank you for your time and valuable feedback! We greatly appreciate your positive feedback about the motivation and technical contributions of our work.
>
> - **Some concepts need to be explained to ease the reading**: Thank you for the constructive suggestion. In the final version, we will ensure to add more comprehensive explanations about ELECTRA and the bi-encoder. Furthermore, we will emphasize the performance drop of the bi-encoder in data scarcity scenarios to highlight our motivation.
> - **Is the proposed model also effective on normal EL datasets**: Theoretically, our proposed model also works in the full-shot setting on normal EL datasets. In our preliminary study, we found that with an increase in training data, the performance of our method improves. We would like to emphasize that our work addresses data scarcity scenarios. Future work may extend the application of PLM-augmented sparse retrievers to the full-shot setting.
>
> If you have any further questions or suggestions, please feel free to let us know.

---

### Official Review · Reviewer_BHTM · 2023-08-04

**Soundness:** 4

**Excitement:**

4: Strong: This paper deepens the understanding of some phenomenon or lowers the barriers to an existing research direction.

**Paper Topic And Main Contributions:**

The paper proposes a way to improve the recall of candidate retrieval stage of entity linking. In addition to the surface text of mentions, it also uses denoised context for candidate retrieval. The context is denoised by extracting most relevant keywords to the mention. A distant supervision approach is applied to extract potential keywords for each mention by using word overlap between the mention’s surface text and entity descriptions. The evaluation is done on a subset of the ZeShEL data set with a few-shot settings. The experimental results suggest significant improvements over traditional BM25 and more recent dense retriever approaches.

**Questions For The Authors:**

1. How much of the left and right context is considered for each mention? Is it the entire text before and after?

**Reasons To Accept:**

* The proposed method is effective and shows a significant boost in performance.

**Reasons To Reject:**

1. The keyword extraction method seems to be expensive as it needs to predict a score for every token. This may increase the latency by a substantial amount as compared to BM25 and dense retrieval.
2. The evaluation is done on a subset of a single data set. More extensive experiments on multiple data sets would help to demonstrate whether this method is effective across data sets/domains.

**Reproducibility:**

3: Could reproduce the results with some difficulty. The settings of parameters are underspecified or subjectively determined; the training/evaluation data are not widely available.

**Reviewer Confidence:**

4: Quite sure. I tried to check the important points carefully. It's unlikely, though conceivable, that I missed something that should affect my ratings.

---

> ### Author Rebuttal · Authors · 2023-08-29
>
> Thank you for the insightful review and valuable feedback! We appreciate your positive feedback and would like to address your concerns:
>
> - **The keyword extraction method seems to be expensive**: While it is true that the token prediction process costs some computational overhead during the inference phase, we would like to emphasize that the time dedicated to token prediction is relatively **small** compared to the overall inference time. For example, in the "Star Trek" domain, token prediction only accounts for approximately 5% of the total inference time.
>
> - **The evaluation is done on a subset of a single data set**: About the experimental setup, our intention was to ensure consistency with prior work by Li et al. (2022) [1], enabling a direct comparison of results. To further demonstrate the effectiveness of our method on other specialized domains, we conduct experiments on NCBI Disease Corpus [2], a widely-used **biomedical** entity linking dataset, under the same few-shot setting as Li et al. (2022). The following table presents the recall@k results on the test set. Our method consistently outperforms the strong baseline, BM25, and the dense retriever, BLINK.
>
> |        | R@10 | R@20 | R@40 | R@50 | R@64 |
> |--------|------|------|------|------|------|
> | ours   | **62.86**| **67.83**| **71.85**| **72.70**| **73.23**|
> | BM25   | 62.33| 65.07| 70.26| 70.90| 71.74|
> | BLINK  | 11.98| 16.04| 20.10| 26.04| 28.02|
>
>
> - **How much of the left and right context is considered for each mention**: The length of mention context is set to 128, while both the left and right contexts have a length of 64. We appreciate your suggestion and will add this in our implementation detail.
>
> If there are any additional questions or concerns, we are happy to engage in further discussions.
>
> References:
>
> [1] Xiuxing Li, Zhenyu Li, Zhengyan Zhang, Ning Liu, Haitao Yuan, Wei Zhang, Zhiyuan Liu, Jianyong Wang. 2022. Effective Few-Shot Named Entity Linking by Meta-Learning. https://arxiv.org/abs/2207.05280
>
> [2] Rezarta Islamaj Doğan, Robert Leaman, Zhiyong Lu. 2014. NCBI Disease Corpus: A Resource for Disease Name Recognition and Concept Normalization. https://www.ncbi.nlm.nih.gov/pmc/articles/PMC3951655/

---

### Official Review · Reviewer_2Pme · 2023-08-05

**Soundness:** 4

**Excitement:**

4: Strong: This paper deepens the understanding of some phenomenon or lowers the barriers to an existing research direction.

**Paper Topic And Main Contributions:**

This paper proposes a PLM-augmented sparse retriever and shows it performs much better than BM25 and dense retrievers in the setting of few-shot entity linking.

**Reasons To Accept:**

The proposed method is interesting and makes a lot of sense to me. The presentation is very clear and easy to follow. The improvement over other methods is big in the experiments.

**Reasons To Reject:**

There is only one dataset, but I'll not consider this a big issue for a short paper.

**Reproducibility:**

4: Could mostly reproduce the results, but there may be some variation because of sample variance or minor variations in their interpretation of the protocol or method.

**Reviewer Confidence:**

3: Pretty sure, but there's a chance I missed something. Although I have a good feel for this area in general, I did not carefully check the paper's details, e.g., the math, experimental design, or novelty.

---

> ### Author Rebuttal · Authors · 2023-08-29
>
> Thanks for your time and constructive feedback! We appreciate your positive feedback about our technical contributions and strong performance.
>
> **Regarding the experimental setup,** our intention was to maintain consistency with prior research efforts, specifically the work of Li et al. (2022) [1], which focused on four specialized domains within ZESHEL. This allows for a direct comparison of results.
>
> To further validate the efficacy of our method, we conduct experiments in another specialized domain, i.e. **biomedical**. We utilize the NCBI Disease Corpus [2], a widely-used biomedical entity linking dataset. Employing the same few-shot setting, we randomly select 50 samples from the training set and 50 samples from the development set. The following table shows the recall@k results on the test set. Notably, our method consistently outperforms the strong baseline, BM25.
>
>
> |        | R@10 | R@20 | R@40 | R@50 | R@64 |
> |--------|------|------|------|------|------|
> | ours   | **62.86**| **67.83**| **71.85**| **72.70**| **73.23**|
> | BM25   | 62.33| 65.07| 70.26| 70.90| 71.74|
> | BLINK  | 11.98| 16.04| 20.10| 26.04| 28.02|
>
>
> Please let us know if you have any additional concerns after you read our response.
>
> References:
>
> [1] Xiuxing Li, Zhenyu Li, Zhengyan Zhang, Ning Liu, Haitao Yuan, Wei Zhang, Zhiyuan Liu, Jianyong Wang. 2022. Effective Few-Shot Named Entity Linking by Meta-Learning. https://arxiv.org/abs/2207.05280
>
> [2] Rezarta Islamaj Doğan, Robert Leaman, Zhiyong Lu. 2014. NCBI Disease Corpus: A Resource for Disease Name Recognition and Concept Normalization. https://www.ncbi.nlm.nih.gov/pmc/articles/PMC3951655/

---

### Meta-Review · Area_Chair_PeEy · 2023-09-19

**Recommendation:** 4

**Metareview:**

The paper presents a method for enhancing the recall of the candidate retrieval phase in the entity linking process. It introduces a keyword extractor based on Pre-trained Language Models (PLM) to create a more refined context and utilizes the BM25 retrieval model for obtaining candidate entities.

---

### Decision · Program_Chairs · 2023-10-07

**Decision:**

Accept-Main

**Comment:**

The paper presents a method for enhancing the recall of the candidate retrieval phase in the entity linking process. It introduces a keyword extractor based on Pre-trained Language Models (PLM) to create a more refined context and utilizes the BM25 retrieval model for obtaining candidate entities.